# Research on the Influence of Vehicle Speed on Safety Warning Algorithm: A Lane Change Warning System Case Study

**DOI:** 10.3390/s20092683

**Published:** 2020-05-08

**Authors:** Rui Fu, Yali Zhang, Chang Wang, Wei Yuan, Yingshi Guo, Yong Ma

**Affiliations:** School of Automobile, Chang’an University, Xi’an 710064, China; furui@chd.edu.cn (R.F.); wangchang@chd.edu.cn (C.W.); yuanwei@chd.edu.cn (W.Y.); guoys@chd.edu.cn (Y.G.); mayong@chd.edu.cn (Y.M.)

**Keywords:** different speed, minimum safety deceleration, naturalistic driving, lane change warning system, comparative verification

## Abstract

Speed has an important impact on driving safety, however, this factor is not included in existing safety warning algorithms. This study uses lane change systems to study the influence of vehicle speed on safety warning algorithms, aiming to determine lane change warning rules for different speeds (DS-LCW). Thirty-five drivers are recruited to carry out an extreme trial and naturalistic driving experiment. The vehicle speed, relative speed, relative distance, and minimum safety deceleration (MSD) related to lane change characteristics are then analyzed and calculated as warning rule characterization parameters. Lane change warning rules for a rear vehicle in the target lane under four-speed levels of 60 ≤ *v* < 70 km/h, 70 ≤ *v* < 80 km/h, 80 ≤ *v* < 90 km/h, and *v* ≥ 90 km/h are established. The accuracy of lane change warning rules not considering speed level (NDS-LCW) and ISO 17387 are found to be 87.5% and 79.8%, respectively. Comparatively, the accuracy rate of DS-LCW under four-speed levels is 94.6%, 93.8%, 90.0%, and 92.6%, respectively, which is significantly superior. The algorithm proposed in this paper provides warning in the lane change process with a smaller relative distance, and the accuracy rate of DS-LCW is significantly superior to NDS-LCW and ISO 17387.

## 1. Introduction

Speed is an important parameter affecting driving safety, with the collision rate and the severity of all road types rising with an increase of speed [1,2]. In the process of driving, speed has an impact on the driver’s psychology and horizon [3,4]. Speed also has a significant impact on all types of collision warning systems. For example, Yamamoto found that the safe distance model is not only related to headway, but also speed, so they established a safe distance model of various speed [5]. Kim found that 70% of urban road collision accidents occur when the speed is slower than 30 km/h, and developed a low-speed short-distance collision warning system [6]. Lane change is considered as one of the most dangerous driving behaviors because drivers have to deal with the traffic conflicts on both the current and target lanes [7]. In the process of the lane change, the driver’s subjective safety lane change conditions vary at different speeds. Compared with low speed, the driver requires a longer relative distance and time to collision (TTC) with other vehicles to meet the needs of safe lane change at high speed.

According to European statistics, traffic accidents caused by lane change account for approximately 4–10% of total traffic accidents [8,9]. Research shows that if the driver receives a warning signal before 0.5 s of collision risk, 50% of the collision accidents can be avoided and if they receive a warning before 1 s, 90% of the accidents can be avoided [10]. Lane change warning systems (LCW) are gradually being developed and popularized in vehicle active safety systems [11,12]. By monitoring the surrounding vehicles and issuing collision warnings to the driver in advance, the system reduces the risk of potential conflict, improves land change safety, and reduces the driver’s operating load [13,14,15]. According to the collected road and traffic information, current LCWs generally utilize the safety lane change evaluation algorithm to determine whether the driver will encounter danger during the lane change process. When determining the danger, the system will provide a warning to the driver so that drivers can take certain measures to avoid danger [16,17,18]. However, the existing LCWs do not adequately consider the impact of speed on the warning threshold. The widely used ISO 17387 LCW algorithm only considers the relative speed, excluding the speed. Under the same relative speed, taking TTC 3 s as an example, if the vehicle is running at 40 km/h or 120 km/h, the driver’s risk perception will be significantly different. The higher the speed is, the more cautious the driver tends to be, and lane change decisions will be different. Therefore, using ISO 17387 will create various problems. When the speed is different, the driver’s lane-changing behavior characteristics also vary. The faster the speed is, the greater the driver’s mental load is, and the stronger the sense of stress is, so the requirements for lane change safety will be higher [19]. If the same warning rules are adopted at different speeds, the probability of false alarm will be increased and the acceptance of LCW by drivers will be reduced. Therefore, exploring the DS-LCW algorithm is of great significance to improve driving safety.

The driving characteristics and psychological load of drivers vary at different speeds. For safety reasons, the warning value set provided by the evaluation algorithm is relatively conservative. However, if the warning value is too conservative and the impact of speed on it is not considered, it will increase the false alarm rate of LCW under different driving speeds and interfere in the normal driving habits of drivers. A balance must be determined when setting the warning value, which not only avoids all possible collisions but also does not interfere with the normal operation of a driver. The setting of this value is particularly important for the practicability and accuracy of LCW [20]. Many LCWs use TTC as the warning parameter [21,22]. Dijck et al. determined that the safety limit that a driver can accept subjectively is TTC 3 s or a 5 m distance from the target vehicle [23]. Alternatively, Hirst et al. reported that the safety limit accepted by the driver could be TTC 4 s [24]. However, an increasing number of studies have found that a single warning threshold cannot adapt to the difference in subjective driver perception of environmental conditions in the actual lane change process. Therefore, in recent years, researchers have sought to improve driver acceptance of LCW by distinguishing a warning level and determining the optimum warning threshold [25,26]. Lee et al. divided the emergency degree of lane change into four stages: TTC > 5.5 s, 3.0 s < TTC ≤ 5.5 s, TTC ≤ 3.0 s, and collision [27]. Yong et al. classified lane change safety into three levels and determined TTC thresholds of 3.0 s and 5.5 s [28]. The TTC can comprehensively reflect the distance and relative speed between two vehicles, and accurately express the dynamic characteristics of the surrounding vehicles and ego-vehicle. However, when the relative speed is close to 0 and the distance is small, TTC cannot correctly reflect the safety of the lane change. Therefore, it is not accurate to use TTC as a single index to evaluate the safety of lane changes. From the driver’s subjective point of view, the most direct parameters to evaluate the danger degree of surrounding vehicles to the host vehicle are speed, distance, and relative speed. Based on the approximate judgment of speed, relative speed, and relative distance, the driver can comprehensively evaluate whether the traffic environment meets the lane change conditions, to decide whether to take lane change operation.

In view of the problem that the single warning threshold of the existing LCW is fixed, which results in early warning at low speed and untimely warning at high speed, this paper takes the speed, relative distance, and relative speed and minimum safety deceleration (MSD) as the warning indicators. A DS-LCW for the rear vehicle in the target lane is then presented. Compared with ISO 17387 and NDS-LCW, it is found that DS-LCW proposed in this paper is more consistent with the driving habits of domestic drivers, reduces the rate of false alarm and false negatives, and improves driver acceptance of LCW.

## 2. Materials and Methods

### 2.1. Apparatus

The test platform used in the experiments was a 2008 Volkswagen Touran, equipped with a variety of data acquisition instruments (Figure 1). The instruments predominantly included two-millimeter-wave radars for measuring the relative speed, relative distance, and relative angle between the test vehicle and surrounding vehicles. The range of millimeter-wave radar is 0.5–200 m, the horizontal measurement angle is ±45°, and the resolution of angle is 0.5°; a video monitoring system for collecting the head motion of drivers, operational behavior, and driving environment; a lane mark recognition system for measuring the lateral distance between the wheels on both sides of the vehicle and the lane marking line, the output frequency of this system is 10 Hz, the measurable range is ±635 cm, and the measurement accuracy is 5 cm; a CAN acquisition card for obtaining the driving parameters of the test vehicle, including vehicle speed, steering wheel angle, etc. The measured accuracy of steering wheel angle is about 1.4°, and that of vehicle speed is 0.01 km/h; a wireless button located on the left side of the steering wheel for recording the lane change extreme moment.

### 2.2. Driver and Driving Route

In this study, 35 licensed drivers (9 female, 26male) were selected to carry out two experiments: a rear extremity test for lane change and a naturalistic driving experiment. The age of the subjects ranged from 25 to 52 years old, with an average age of 37.6 years (SD = 6.92). Their driving experience ranged from 2 to 30 years, with an average value of 13.2 years, and a standard deviation of 8.12.

The rear extremity test mainly focused on the process of lane change with a fast-approaching rear vehicle, and the test vehicle is in front of the rear vehicle. As shown in Figure 2, *M* is the test vehicle and *F_d_* is the fast-approaching rear vehicle. In the actual lane change, when a fast coming rear vehicle in the target lane approaches the ego-vehicle gradually, lane change will gradually change from a safe state to a dangerous state. The turning point of these two states is called the latest safe lane change moment. In this paper, the moment was defined as extreme time, which was described as an inevitable conflict with *F_d_* if *M* implemented a lane change after that moment. 

The purpose of this test is to obtain the driving data of vehicle *F_d_* and *M* at the extreme time, using them to analyze the deceleration of vehicle *F_d_*, which can provide the basis for setting the LCW threshold. During the test, two experimenters accompanied. They sat in the front seat and the back seat respectively. The assistant experimenter was responsible for assisting the subjects to observe the traffic scene in front of the test car. The back experimenter was responsible for observing whether there was a fast coming vehicle from behind in the target lane and reminding the subjects to look at the rear-view mirror. After receiving the signal from the experimenter, the driver of *M* observes repeatedly *F_d_* from the rear-view mirror and judges the critical moment that *M* can safely change to the left lane. Because of the danger inherent in these extremity tests, the driver would press a wireless button installed on the left side of the steering wheel (shown in Figure 1) to indicate the lane change process instead of actually changing lanes when they were aware of the extreme time. The data at the critical moment for vehicle *M* and *F_d_* were recorded as the extremity test data. To reduce the driving load, during the test, vehicle *M* ran at the fixed speed with the aid of a constant speed cruise system, and the speed was set at 60, 70, 80, and 90 km/h, respectively.

The naturalistic driving experiment was carried out on a different test route and included ordinary road, urban expressway, and highway, with a round trip mileage of about 100 km. This test is aimed at collecting lane change behavior data in the actual road, to analyze the characteristics of lane-change parameters, and to verify the effectiveness of lane change warning rules. The driver was informed of the route in advance of the naturalistic driving experiment. During the test, the driver was asked to drive under their driving habits, without restrictions or requirements on speed, operation habits, and vision. The data and video were collected throughout the entire process.

### 2.3. Process of Data Acquisition

Compared with other vehicles, in the process of lane change, the driver’s judgment of the rear vehicle must be completed with the aid of the rear-view mirror, which is prone to inaccurate judgment, while the existence of visual blind area further complicates judgement [29]. Therefore, only the safety of vehicle *F_d_* is analyzed in this paper as it has the greatest impact on the safety of a lane change.

Extremity test data acquisition: The target vehicle was confirmed using a video recorded by the multi-channel on-board video monitoring system, and driving parameters such as the speed of vehicle *M*, relative speed, and the relative distance between *M* and *F_d_* at the extreme time were extracted. According to this method, 1856 valid data were extracted.

Safe lane change and unsafe lane change data were screened based on the naturalistic driving data. The scene existing for *F_d_* in the process of lane change was selected from the driving video, then the speed of *M*, relative speed, relative distance, angle, and other information between the *M* and the *F_d_* was extracted as the basic data for establishing warning rules.

Data screening for safe lane change: The successful lane change process was defined as safe lane change. Warning rules were established for selective lane change behavior in this paper. The following principles were followed to screen the safe lane change data:Mandatory lane change data were eliminated;Continuous lane change data intending to achieve the desired speed was eliminated;When no other vehicles were included within the radar detection range when changing lanes, the data were eliminated because of the low probability of danger and no need for a warning.Low-speed lane change data of congested road sections were eliminated.

Based on the above screening conditions, combined with the video, 2519 safe lane changes of *F_d_* were selected, among them, the data from 2011 reflected that *F_d_* was faster than *M*.

Data screening for unsafe lane change: Unsafe lane change means that the driver has the intention of lane change, but does not perform the lane change because of some unsafe factors. As it is impossible to accurately understand whether the driver has the intention of a lane change, the selected videos met the following conditions:The front vehicle was in the existing main lane and its speed was slow, affecting the normal driving of vehicle *M*;The driver observed the rear-view mirror and found that *F_d_* was approaching *M* quickly or *F_d_* was relatively close to *M*, threatening the lane change behavior of *M*;After *M* was overtaken by *F_d_*, *M* changed lanes after a short time.

According to the above conditions, a total of 1645 unsafe lane changes were selected, of which 1598 were canceled because of the fast-approaching of *F_d_*, and 47 were canceled because of the short distance between *F_d_* and *M*.

### 2.4. Characteristic Analyses of Lane Change Parameter

To explore the influence of speed on the characteristic parameters of the lane change, the speed is divided into four levels according to the test data: 60 ≤ *v* < 70, 70 ≤ *v* < 80, 80 ≤ *v* < 90, and *v* ≥ 90, with the unit of km/h. The changing trend of the TTC, relative distance, and relative speed between *M* and *F_d_* with speed during the safe lane change is then analyzed, as shown in Figure 3, Figure 4 and Figure 5. In this paper, the relative speed *v_r_* (m/s) is defined as the difference between the speed of *F_d_* and that of *M*. That is, *v_r_* > 0: the speed of *F_d_* is higher than that of *M*, and the distance between the two vehicles is decreasing; *v_r_* < 0: the opposite phenomenon of above. As illustrated in Figure 3 and Figure 4, the mean value and percentile value of TTC and relative distance rise with an increase in the speed of *M*. As shown in Figure 5, with an increase in the speed of *M*, the relative speed decreases, indicating that the driver is more cautious when changing lanes at high speed. The changing trend of lane change parameters is consistent with the driver’s subjective feelings. With the increase of speed, the driver’s mental load will increase, they can feel a stronger sense of stress, and higher lane change requirements are enacted.

### 2.5. Establishing DS-LCW

In the past, TTC was predominantly utilized as the indicator of lane change warning rules and the influence of speed on warning threshold was not considered. As shown in the previous section, speed has a significant influence on lane change parameters, especially TTC. If the influence of speed is not considered, using only TTC as an indicator will result in early warnings at low speed and untimely warnings at high speed. This high false alarm rate is highly unfavorable to the promotion and use of LCW. Therefore, it is necessary to establish a DS-LCW. In this section, the speed, MSD of *F_d_*, and relative distance are used as indicators to establish DS-LCW.

When the speed of *F_d_* is higher than that of *M* ( i.e., *v_r_* > 0), according to the lane change warning model based on driver perception characteristics established in our previous research [20,30], when *F_d_* approaches *M* quickly if *M* is perceived to have lane change intention or the body of *M* has lateral displacement, the driver of *F_d_* will slow down to maintain a safe distance. According to the previous model, the MSD of *F_d_* during the extremity test is calculated according to Equation (1).
(1)$$a=\frac{v_{r}^{2}}{2(d-D-v_{r}T)}$$
where *a* is MSD of *F_d_*, *v_r_* is the relative speed between *F_d_* and *M*, *d* is the relative distance, *D* is the minimum safety distance between two vehicles, and *T* is the reaction time of the driver for *F_d_*. Both *v_r_* and *d* use data obtained from the radar and the CAN acquisition card. Numerous scholars have studied the response time in the process of braking behavior, finding that the deceleration response time is 1 s [31,32]. For the driver’s response time of *F_d_* in this paper, *T* refers to the previous research and is 1 s. According to the research of Sultan [33], when the relative speed of two vehicles is in the range of *R* = [−1.5 m/s, 1.5 m/s], it is assumed that they tend to maintain the appropriate safety distance. The relative distance when the relative speed is located in *R* is statistically analyzed using the data of safe lane change. The results show that the maximum value is 132.99 m, the minimum value is 4.58 m, the average value is 26.23 m (SD = 4.35). Therefore, in this paper, *D* is 4.58 m, according to the minimum value.

In the extremity test, the vehicle adopts a constant speed cruise, the percentile of MSD under four speeds is then calculated. As shown in Figure 6, with the increase of speed, the average value of MSD and 25%, 50%, and 75% quantiles of it show a decreasing trend.

As *D* and *T* in Equation (1) are constant, under the determined MSD, the relationship between relative velocity *v_r_* and relative distance *d* is a quadratic function, as shown in Equation (2). The percentile of MSD is taken as the warning threshold in the case of *v_r_* > 0. The data of safe lane change and unsafe lane change in the process of naturalistic driving is then used to evaluate the effectiveness of the warning threshold, as shown in Figure 7. The 50% quantile of MSD obtains a good effect on the sample point partition of safe lane change and unsafe lane change, and the division accuracy reaches 92.75%. Therefore, the 50% quantile of MSD is taken as the warning threshold of lane change when *v_r_* > 0.
(2)$$d=\frac{1}{2a}v_{r}^{2}+v_{r}+4.58$$

As shown in Figure 8, the partition effect of MSD percentiles on the sample point of safe lane change and unsafe lane change is observed when the speed level is not differentiated. Regardless of 25%, 50%, or 75% quantile, the partition effect for two types of data is not significant, which will cause a high false alarm rate or false-negative rate. This phenomenon further demonstrates the necessity of establishing the proposed DS-LCW.

When the speed of *F_d_* is lower than that of *M* (i.e., *v_r_* < 0), the majority of naturalistic driving data is the safe lane change. In actual driving, when the relative distance between *M* and *F_d_* is small, it can easily cause collision accidents. In this case, the driver tends to cancel the lane change operation, so the LCW is highly necessary to warn for short-distance lane change operations and can effectively avoid the occurrence of collision accidents. In Lee’s study, 500 lane change samples were collected and analyzed, and 5% and 25% quantiles of the relative distance, relative speed, and TTC between the test vehicle and surrounding vehicles were taken as lane change warning thresholds [27]. Referring to this study, the 5% quantile of relative distance is used as the warning threshold when *v_r_* < 0. Taking the percentile of safety distance as the warning threshold is based on an assumption of the acceptance and safety of the warning rules by most drivers.

Following this, 1200 data are randomly selected from 2519 safety lane change data, which are divided into the same four-speed levels. The changing trend of the relative distance with speed for 1200 data is shown in Figure 9, the same as Figure 4, the mean value and percentile value of relative distance rise with an increase in the speed of M. This shows that 1200 data have a representative of other drivers in the population to some extent. According to the statistics, 5% quantiles of the relative distance are selected as the warning threshold of *v_r_* < 0, as 4.8 m, 5.0 m, 5.3 m, and 5.5 m, respectively.

Combining the threshold of *v_r_* > 0 and *v_r_* < 0, the DS-LCW are obtained as shown in Table 1.

## 3. Results

### 3.1. Comparison of Warning Rules

International standards and evaluation procedures for lane change collision warning systems have been formulated, among which ISO 17387 is widely used. China has also formulated national standards for LCW regarding ISO standards [34,35]. In this section, the DS-LCW proposed in this paper is compared with NDS-LCW and ISO 17387, to verify its feasibility.

As the ISO 17387 uses different TTC thresholds in different relative speed ranges, it is only related to relative speed and independent of speed. For each relative speed range, the critical TTC threshold and the relative distance at the initial time of lane change are respectively shown in Table 2.

According to the established method of the warning rules in this paper, for the NDS-LCW, the 50% quantile of MSD is 1.73 m/s^2^, and the 5% quantile of relative distance is 5.0 m. The three warning rules are compared and verified using the data of safe lane change and unsafe lane change of naturalistic driving experiment, as shown in Figure 10 and Figure 11. A comparison between DS-LCW and ISO 17387 is provided in Figure 10. Taking the 60 ≤ v < 70 km/h (Figure 10(a)) as an example to illustrate the warning rules, the red dot represents the sample point of the unsafe lane change, the green pentagram represents the sample point of the safe lane change, and the black curve represents DS-LCW. The upper left area of it is the safety area, and the lower right area is the warning area. The blue curve is the ISO 17387, the lower right area belongs to its warning area, and the rest belongs to the safety area. A comparison between NDS-LCW and ISO 17387 is shown in Figure 11. It can be observed from the figure that the accuracy of DS-LCW is higher than that of NDS-LCW and ISO 17387 in dividing the data of safe lane change and unsafe lane change.

The feasibility of the warning rules can be verified according to the acceptance degree of the driver. The sample collected in this experiment is the lane change data of the driver under the normal driving state. If the sample points fall into the warning area, it means that the warning rules will alert the driving state corresponding to the data points. Table 3 shows the quantity distribution and proportion of the F_d_ lane change samples in the warning areas of DS-LCW, NDS-LCW, and ISO 17387. As illustrated in the table, for the safety lane change samples, the proportion distributed in the warning area of DS-LCW is larger than that of ISO 17387, which corresponds to the warning rule line of DS-LCW in Figure 10 and is above that of ISO 17387. For the unsafe lane change samples, the proportion distributed in the ISO 17387 warning area is much larger than that of DS-LCW, which shows that the warning rules set in this paper are more in line with the driver’s danger judgment standards.

Compared with NDS-LCW, for the DS-LCW proposed in this paper, the distribution proportion of safe lane change samples in the warning area and the proportion of unsafe lane change samples in the safe area are lower at each speed level. Once again, this illustrates the impact of speed on the LCW and the necessity of distinguishing speed to establish lane change warning rules. Overall, although the false alarm rate of the warning rules proposed in this paper is slightly higher than that of ISO 17387 for safe lane change samples, the false-negative rate of the DS-LCW is much lower than that of ISO 17387 for unsafe lane change samples. Besides, this work includes warning lane change samples with smaller relative distance, which makes the warning rules more accurate to precisely provide a warning to drivers in unsafe lane change driving states.

### 3.2. Verification of Warning Rules

Signal detection theory (SDT) is utilized to further verify the effectiveness of the DS-LCWs. This technology has been widely used in the optimal threshold of human perception [37,38,39]. The lane change decision matrix is provided in Table 4. When the system sends a safety signal and the driver changes lane, this is called “hit”; when the system sends a safety signal but the driver cancels the lane change operation, this is called “false negative”; when the system sends a warning signal but the driver changes lane, this is called “false alarm”; when the system sends a warning signal and the driver cancels the lane change operation, this is called “correct rejection”.

Both “hit” and “correct rejection” are correct signals, and the system accuracy is defined as *P*. *P*, *P_FN_*, and *P_FA_* are used to evaluate the effectiveness of the warning rules. The larger the *P*, the better the warning rules are, which is more close to the decision-making of the driver’s subjective lane change operation judgment.
(3)$$P=1-\frac{N_{FA}+N_{FN}}{N_{S}+N_{U}}$$
(4)$$P_{FA}=\frac{N_{FA}}{N_{S}}$$
(5)$$P_{FN}=\frac{N_{FN}}{N_{U}}$$
where *N_S_* is the total number of safe lane change samples, *N_U_* is the total number of unsafe lane change samples, *N_FN_* is the number of false negatives, and *N_FA_* is the number of false alarms. The above three quantities are calculated for ISO 17387, NDS-LCW, and DS-LCW, as shown in Table 5. It is found that the warning accuracy of DS-LCW for different speeds is 94.6%, 93.8%, 90.0%, and 92.6% respectively, and the average value is 13% higher than that of ISO 17387 and 5.3% higher than that of NDS-LCW. The results illustrate that the DS-LCW possess higher practicability and are closer to the daily lane change habits of Chinese drivers. The false alarm rate of DS-LCW is 7.4% for safe lane change samples, which is higher than that of ISO 17387, but the false-negative rate of unsafe lane change samples is 7.1%, which is far lower than 51.3% of ISO 17387. This shows the risk judgment of the warning rules proposed in this paper corresponds more accurately with the driver’s subjective responses.

## 4. Discussion

In the process of driving, speed has a significant influence on the driver’s dynamic vision, horizon, and mental state. With the increase of speed, the dynamic vision of the driver decreases, the fixation point moves further away, surrounding objects became blurred, and the dynamic horizon is significantly narrowed. When driving at high speed, the attention of the driver is focused on the traffic conditions of the road ahead and the psychological load is high. The estimation of speed and distance almost entirely depends on the subjective feeling of the driver. According to the experiment of American scholar in 1972, when driving in the same direction, the driver’s judgment of headway is smaller than the actual distance, and the higher the speed is, the greater the difference between the judged headway and the actual distance is. In the lane change process, with the increase of speed, the difference between the driver’s judgment of the distance of the rear vehicle and the actual distance increases. Therefore, when changing lanes at high speed, the driver will prefer to choose a larger relative distance to ensure safety. In the same way, when driving at high speed, the driver experiences a strong sense of psychological pressure, the surrounding traffic environment provides less stimulation, judgment ability is weakened, the response time becomes longer, and the TTC of the lane change is also extended. Therefore, speed has a large influence on the LCW threshold setting.

However, as TTC is often used as the warning index in existing LCWs, the influence of speed on the threshold of LCW has not been studied. This paper analyzed the changing trend of lane change parameters with speed based on the data of naturalistic driving, finding that with the increase of speed, TTC displays an obvious increasing trend. Therefore, using a fixed TTC threshold at all speeds will result in a high false alarm rate of LCW and reduce the practicability and effectiveness of the system. To solve the problem of the existing LCW lacking a discussion of speed, based on the extremity test and naturalistic driving experiment, lane change data including the vehicle *F_d_* was selected to establish a DS-LCW with the threshold indicators of speed, relative distance, and MSD. By comparing DS-LCW with NDS-LCW and ISO 17387, the effectiveness and practicability of the warning rules proposed in this paper were verified.

When *v_r_* > 0, a 50% quantile of MSD was selected as the warning threshold. For four speed levels, the warning thresholds are 2.47, 1.77, 1.29, and 1.15 m/s^2^, respectively. With the increase of speed, the warning threshold decreases. As shown in Equation (2), MSD is located on the denominator of quadratic, and the quadratic coefficient increases with the increase of speed. According to the characteristics of a quadratic function, it can be seen that the larger the quadratic coefficient, the smaller the curve opening. Therefore, as shown in Figure 7 and Figure 10, the warning line is closer to the longitudinal axis as speed increases. From the view of TTC, in the four figures shown in Figure 10, a point with the same relative speed and relative distance on the warning line were taken and used as a tangent. The slope is the TTC under the state of a lane change. With the increase of speed, the slope increases, meaning that the TTC increases with the increase of speed. This conclusion is consistent with the changing trend of lane change characteristic parameters with speed analyzed in Section 3.

When *v_r_* < 0, a 5% quantile of relative distance was used as a warning threshold to warn the lane change process with a smaller relative distance. The warning thresholds of four-speed levels are 4.8 m, 5.0 m, 5.3 m, and 5.5 m, respectively. Similarly, with the increase of speed, the warning threshold shows an increasing trend, which also is consistent with the results in Section 3.

With the help of SDT, the DS-LCW was compared with NDS-LCW and ISO 17387 based on the data of the naturalistic driving experiment. It was found that the warning accuracy of NDS-LCW is 87.5%, and under four-speed levels, that of ISO 17387 is 84.1%, 83.9%, 73.8%, and 77.3%, respectively. After distinguishing speed, the warning accuracy rate of the DS-LCW is 94.6%, 93.8%, 90.0%, and 92.6% in the four-speed levels, respectively. 

Considering the adaptability of DS-LCW to different vehicle types and driving environments, one of the warning indicators in this paper is the minimum safety deceleration of the rear vehicle in the target lane, i.e., MSD mentioned in this paper. Deceleration is used to make lane changing safety decisions, for different vehicle types, the deceleration characteristics of the vehicle themselves are different, so when setting the warning rules for a certain vehicle type, it is necessary to take the deceleration characteristics of the vehicle into account. For example, the maximum deceleration value of a small car is usually higher than that of a large size car, and the braking response time of a small car is also short. In terms of driving environment, DS-LCW in this paper mainly considered the speed of ego-vehicles and the surrounding specific vehicles (i.e., the fast-approaching rear vehicle in the target lane). As long as these two conditions are met, the lane change warning rules are applicable in any driving environment such as urban road, highways, and etc.

## 5. Conclusions

The average value of the warning accuracy rate for DS-LCW was 5.3% higher than that of NDS-LCW and 13% higher than that of ISO 17387. The findings show that under the premise of considering the safety of lane changes, the DS-LCW improves the accuracy and practicability of the system, and solves the problem of high false alarm rate caused by fixed warning threshold at different speeds.

### 5.1. Implications for Practice

In practical application, once the system recognizes the driver’s lane change intention, the LCW will immediately collect the kinematic parameters of the host vehicle and vehicle *F_d_*, and correlate the different warning rules with the current speed. The LCW will judge the positive and negative of the relative speed first. If it is positive, the MSD of vehicle *F_d_* at the current time will be calculated according to Equation (1). If it is greater than the warning threshold, the driver will be notified to be careful of potential danger. If it is less than the threshold, the warning system will not respond. When it is determined that the relative speed is negative, the current relative distance will be compared with the threshold value. If it is greater than the threshold value, it is in a safe state, and if it is less than that, the system will supply a danger warning to the driver. In fact, an appropriate warning threshold would directly affect the driver’s acceptance of the LCW. For future automatic driving, DS-LCW proposed in this paper also has some application enlightenment. Establishing LCW by distinguishing speed, it can improve the lane change safety of automatic driving vehicles.

### 5.2. Limitations and Future Research

The contribution of this work was the establishment of DS-LCW for rear vehicles in a target lane to provide warning during lane change processes with smaller relative distance. The percentile of warning index was selected as the warning threshold, although the result showed that had a good effect for distinguishing the data of safe lane change and unsafe lane change, limited by the sample size, the percentile has a certain degree of error. Furthermore, in the extremity test, the extreme time was determined by the subjective feelings of drivers. Owing to the influence of driving style, for conservative and aggressive drivers, the critical moment they considered for safe lane change will be significantly different, this will have an impact on the computation of MSD, to further influence the warning threshold.

There are two points to be improved in future research. One is to increase the sample size to improve the accuracy of the percentile. The other is to find a more objective test and method of determining the warning threshold to reduce the impact of driving style. For example, based on the obtained data, using machine learning technology to determine the warning threshold will reduce the limitation of threshold determination and make it more adaptive, which will be further considered in our future research. Also, acceptance by drivers of alarm strategy should be investigated, which will further improve the performance of LCW and make it closer to the driver’s habits.

## Figures and Tables

**Figure 1 sensors-20-02683-f001:**
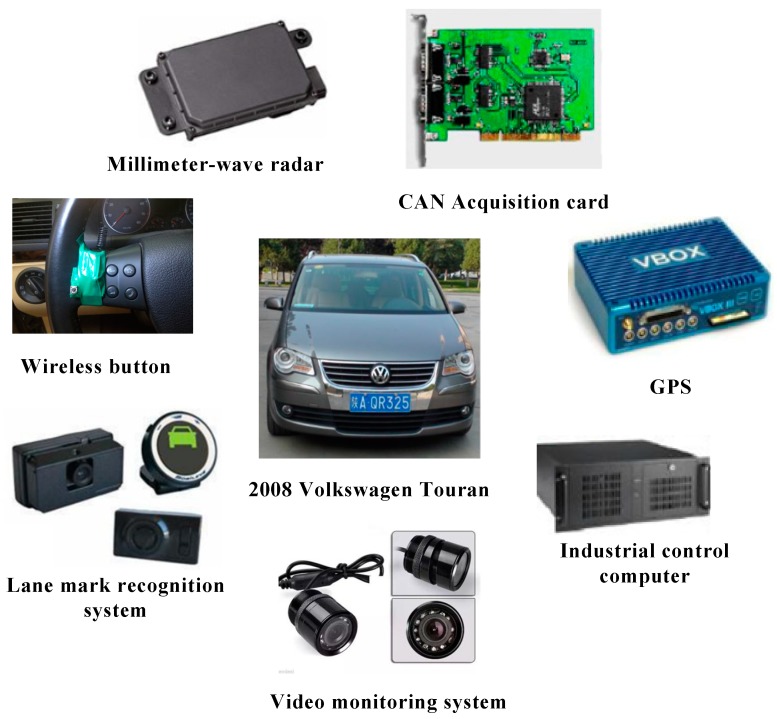
Test platform

**Figure 2 sensors-20-02683-f002:**
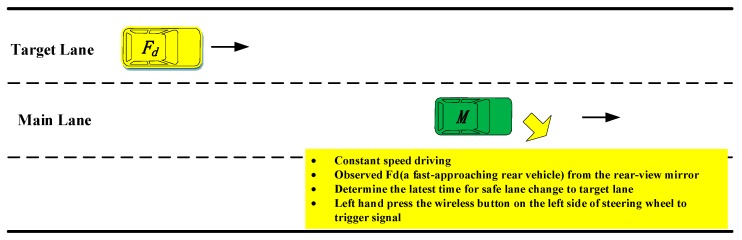
Extremity test.

**Figure 3 sensors-20-02683-f003:**
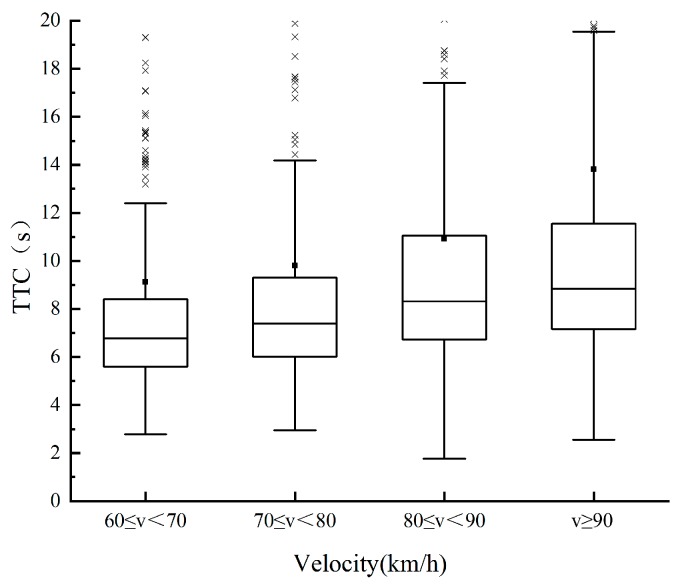
Change trend of TTC.

**Figure 4 sensors-20-02683-f004:**
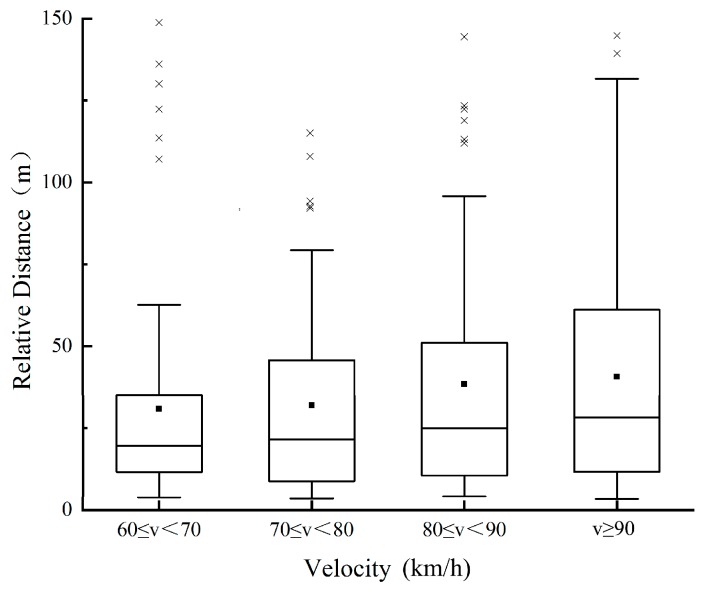
Change trend of relative distance.

**Figure 5 sensors-20-02683-f005:**
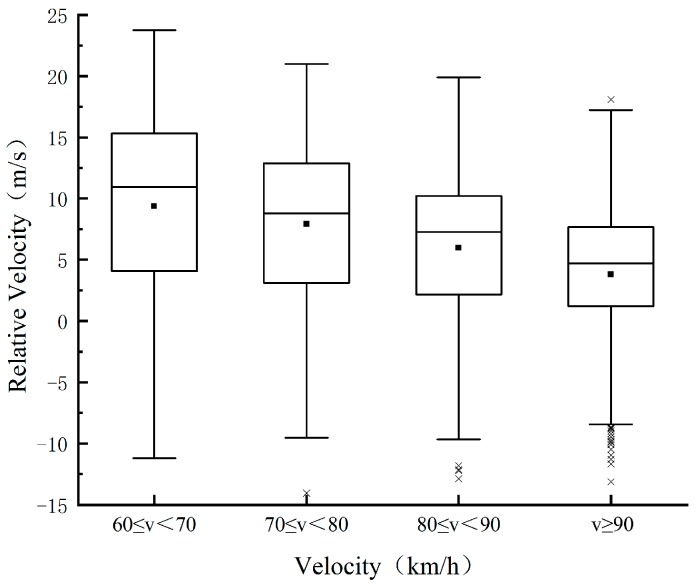
Change trend of relative velocity.

**Figure 6 sensors-20-02683-f006:**
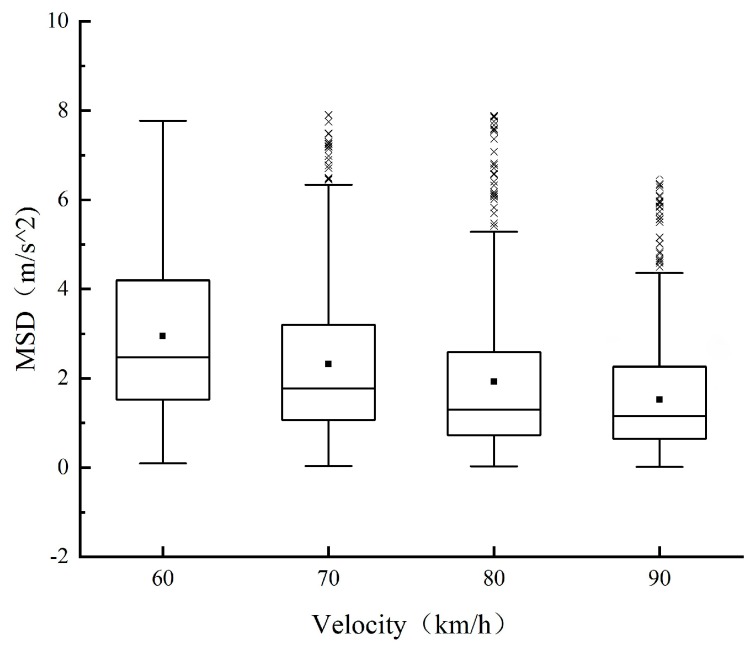
Change trend of minimum safety deceleration (MSD) with speed.

**Figure 7 sensors-20-02683-f007:**
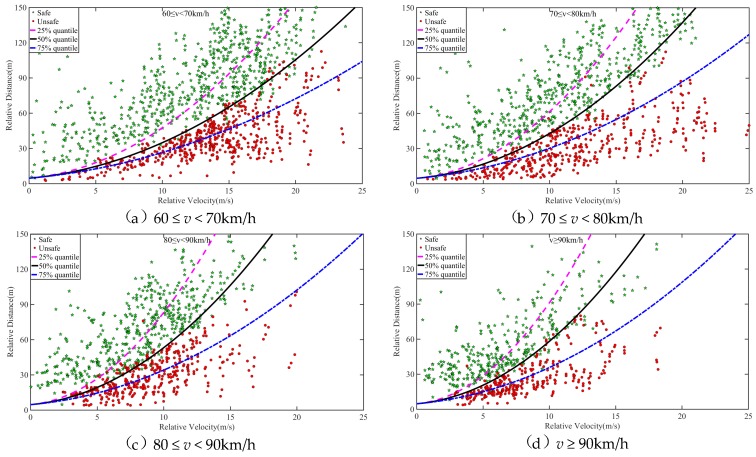
Partition effect of MSD percentiles for the sample point of safe lane change and unsafe lane change.

**Figure 8 sensors-20-02683-f008:**
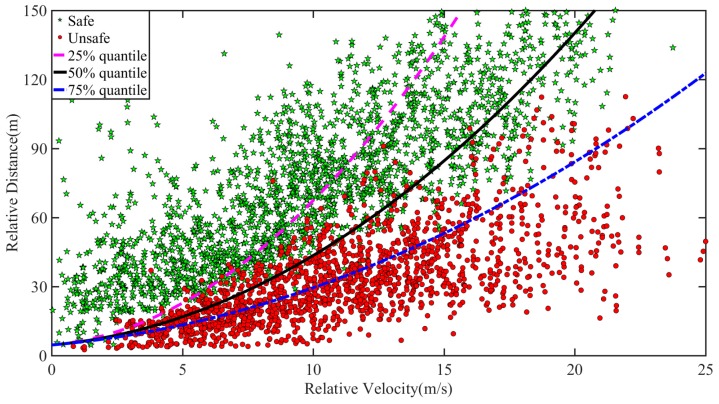
Partition effect of MSD percentiles on the sample point of safe lane change and unsafe lane change when the speed level is not differentiated.

**Figure 9 sensors-20-02683-f009:**
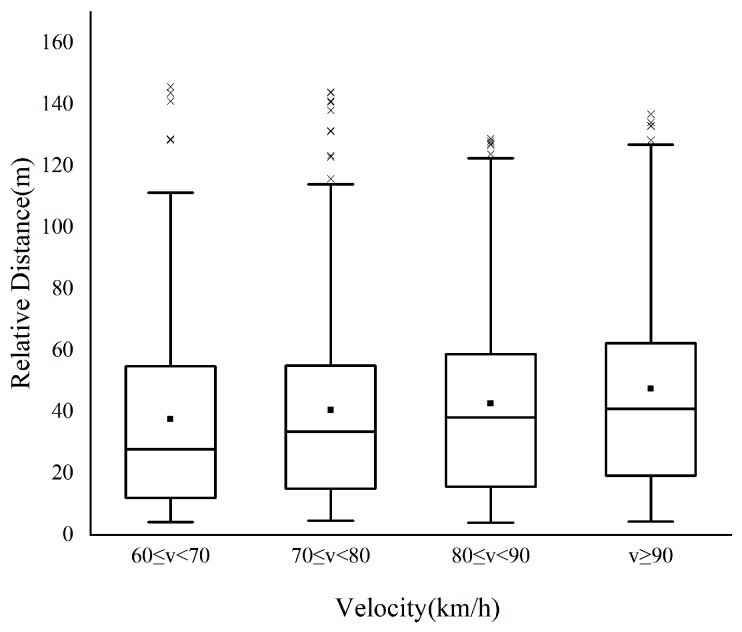
Changing trend of the relative distance with speed for 1200 data.

**Figure 10 sensors-20-02683-f010:**
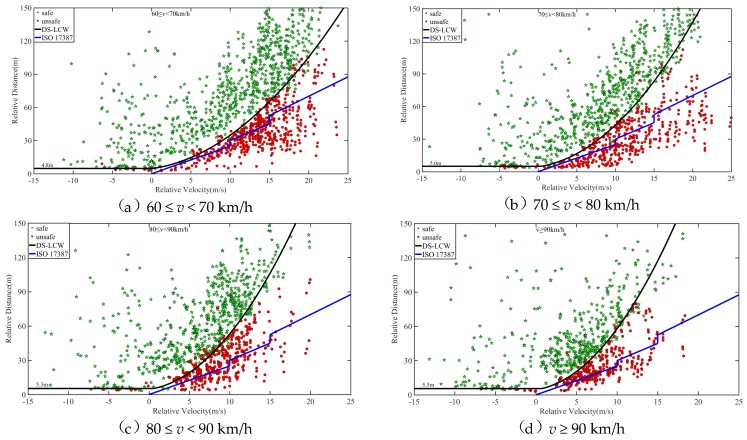
Comparison between DS-LCW and ISO 17387.

**Figure 11 sensors-20-02683-f011:**
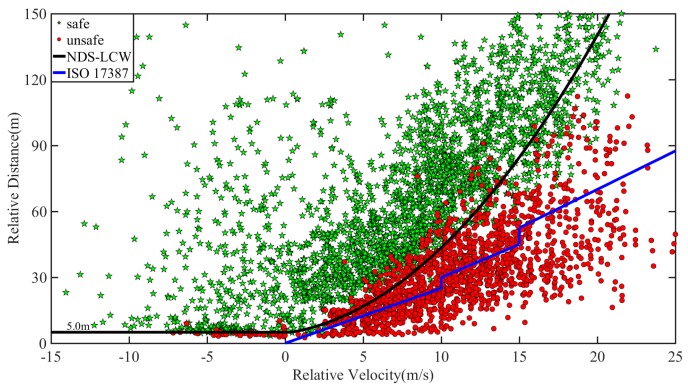
Comparison between NDS-LCW and ISO 17387.

**Table 1 sensors-20-02683-t001:** Lane change warning rules for different speeds (DS-LCW).

	Warning Indicator	60 ≤ *v* < 70 km/h	70 ≤ *v* < 80 km/h	80 ≤ *v* < 90 km/h	*v* ≥ 90 km/h
*v_r_* > 0	50% quantile of MSD (m/s^2^)	2.47	1.77	1.29	1.51
*v_r_* < 0	5% quantile of d (m)	4.8	5.0	5.3	5.5

**Table 2 sensors-20-02683-t002:** ISO 17387 lane change warning rules [36].

TTC(s)	Relative Speed *v_r_* (m/s)	Relative Distance at the Initial Time of Lane Change *d*_0_ (m)
2.5	3	7.5
2.5	5	12.5
2.5	7	17.5
2.5	9	22.5
3	11	33
3	13	39
3	15	45
3.5	17	59.5

**Table 3 sensors-20-02683-t003:** Quantity distribution and proportion of the Fd lane change samples.

	Safe Lane Change Samples					Unsafe Lane Change Samples				
	60 ≤ *v* < 70	70 ≤ *v* < 80	80 ≤ *v* < 90	*v* ≥ 90	NDS-LCW	60 ≤ *v* < 70	70 ≤ *v* < 80	80 ≤ *v* < 90	*v* ≥ 90	NDS-LCW
The number of samples	780	652	618	469	2519	508	443	395	299	1645
The warning area of ISO	0	0	1	0	1	303	267	131	125	826
	0%	0%	0.2%	0%	0.2%	59.6%	60.3%	33.2%	41.8%	50.2%
The safe area of ISO	780	652	617	469	2518	205	176	264	174	819
	100%	100%	99.8%	100%	99.8%	40.4%	39.7%	66.8%	58.2%	49.8%
The warning area of DS-LCW	39	47	51	42	238	477	422	345	284	1363
	5.0%	7.2%	8.3%	9.0%	9.4%	93.9%	95.3%	87.3%	95.0%	82.9%
The warning area of DS-LCW	741	605	567	427	2381	31	21	50	15	282
	95.0%	92.8%	91.7%	91.0%	90.6%	6.1%	4.7%	12.7%	5.0%	17.1%

**Table 4 sensors-20-02683-t004:** Lane change decision matrix.

	Safe Lane Change	Unsafe Lane Change
Safety signal	Hit	False negative
Warning signal	False alarm	Correct rejection

**Table 5 sensors-20-02683-t005:** *P*, *P_FN_*, and *P_FA_* for the above three lane change rules.

	*P*(%)				*P_FN_*(%)				*P_FA_*(%)			
	60 ≤ *v* < 70	70 ≤ *v* < 80	80 ≤ *v* < 90	*v* ≥ 90	60 ≤ *v* < 70	70 ≤ *v* < 80	80 ≤ *v* < 90	*v* ≥ 90	60 ≤ *v* < 70	70 ≤ *v* < 80	80 ≤ *v* < 90	*v* ≥ 90
ISO 17387	84.1%	83.9%	73.8%	77.3%	40.4%	39.7%	66.8%	58.2%	0%	0%	0.2%	0%
Average	79.8%				51.3%				0.05%			
DS-LCW	94.6%	93.8%	90.0%	92.6%	6.1%	4.7%	12.7%	5.0%	5.0%	7.2%	8.3%	9.0%
Average	92.8%				7.1%				7.4%			
**NDS-LCW**	87.5%				17.1%				9.4%			

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
