# Peer review of "Research on the Influence of Vehicle Speed on Safety Warning Algorithm: A Lane Change Warning System Case Study"

_sensors, 2020, doi:10.3390/s20092683_

Round 1

Reviewer 1 Report

A study about the influence of vehicle speed on safety warning algorithms in lane change systems to determine warning rules for different speeds (DS-LCW) is carried out in this paper and it is the main subject the authors deal with. They propose a DS-LCW approach with accuracy and practicability significantly superior to current NDS-LCW and ISO 17387. Authors argue the existing LCWs (i.e., ISO 17387) do not adequately consider the impact of speed on the warning threshold (it uses TCC –Time to Collision- as a single index to evaluate the safety of lane changes). In this sense, if the same warning rules are adopted at different speeds, the probability of false alarms and false negatives will be increased and the acceptance of LCW by drivers will be reduced. Therefore, they propose to exploring the DS-LCW algorithm to improve driving safety, taking into account the speed, relative distance, and relative speed and MSD -minimum safety deceleration- as the warning indicators. Experimental results are presented to evaluate the performance of the proposed solution (Experiments with 35 licensed drivers are carried out in a rear extremity test for lane change and a naturalistic driving test). Results show that the warning accuracy of DS-LCW in the proposed speeds levels are higher than 90%, and the average value is 13% higher than that of ISO 7387 and 5.3% higher than that of NDS-LCW. However, I think that the authors should make an effort to improve the paper by taking into account the following remarks:

  • Figure 1 should be improved, with more detailed information on larger images with better views of each of the testbed components.

  • Figure 2 should be improved, with more descriptive information that helps to visualize the explanation of the different tests carried out as experiments.

  • The proposed approach in this study should be adapted to different driving environments and also be suitable for any vehicle. So, it is desirable to demonstrate such driving environments adaptability. It is also important to demonstrate the feasibility and its effectiveness with different types of vehicles (dynamics, blind points, size, weight, etc.). Could there be differences in the results or not?

  • Statistical analysis should be carried out to verify that the results obtained from the selected samples are representative of other drivers in the population.

  • The conditions for an approach using machine learning techniques to make the new way of calculating the threshold more adaptive, taking advantage of the acquired data, could be addressed by the authors in the paper as future work.

Reviewer 2 Report

The article describes research results in the field of lane change warning algorithms. The introduction, description of problem provides enough details for the reader to understand the challenges. The article is the continuation of the authors work from [19]. In general, I do think that the authors present an interesting finding and enough novelty.

My questions to the authors:

  1. The article is proposed to be published in Sensors. In the references there is very little articles published from Sensors. Maybe this is not the right journal for publication? The authors themselves have published in Sensors, which should be refenced.
  2. As this Sensors journal, the data management and capture of the data should be much more described. The experiment should include data acquisition details, capture refresh rates, etc. Section 2.1 need extensive expansion.
  3. Test should also be carried out to confirm these findings, not only observation and calculation of MSD.
  4. Is the title correct for table 5? "Table 5. P, PFN, and PFA for the above three lane change rules"

Round 2

Reviewer 2 Report

The authors have edited the article according to the reviewer's comments. The quality of the paper has been increased.